# Strategies to Obtain Designer Polymers Based on Cyanobacterial Extracellular Polymeric Substances (EPS)

**DOI:** 10.3390/ijms20225693

**Published:** 2019-11-14

**Authors:** Sara B. Pereira, Aureliana Sousa, Marina Santos, Marco Araújo, Filipa Serôdio, Pedro Granja, Paula Tamagnini

**Affiliations:** 1i3S - Instituto de Investigação e Inovação em Saúde, Universidade do Porto, Rua Alfredo Allen, 208, 4200-135 Porto, Portugal; 2IBMC - Instituto de Biologia Celular e Molecular, Universidade do Porto, Rua Alfredo Allen, 208, 4200-135 Porto, Portugal; 3INEB - Instituto de Engenharia Biomédica, Universidade do Porto, Rua Alfredo Allen, 208, 4200-135 Porto, Portugal; 4ICBAS - Instituto de Ciências Biomédicas Abel Salazar, Rua de Jorge Viterbo Ferreira 228, 4050-313 Porto, Portugal; 5FEUP - Faculdade de Engenharia, Departamento de Engenharia Metalúrgica e Materiais, Universidade do Porto, Rua Dr. Roberto Frias, 4200-465 Porto, Portugal; 6FCUP - Faculdade de Ciências, Departamento de Biologia, Universidade do Porto, Rua do Campo Alegre, Edifício FC4, 4169-007 Porto, Portugal

**Keywords:** EPS-based biomaterials, cyanobacteria, designer biopolymers, extracellular polymeric substances (EPS), metabolic engineering, polymer functionalization

## Abstract

Biopolymers derived from polysaccharides are a sustainable and environmentally friendly alternative to the synthetic counterparts available in the market. Due to their distinctive properties, the cyanobacterial extracellular polymeric substances (EPS), mainly composed of heteropolysaccharides, emerge as a valid alternative to address several biotechnological and biomedical challenges. Nevertheless, biotechnological/biomedical applications based on cyanobacterial EPS have only recently started to emerge. For the successful exploitation of cyanobacterial EPS, it is important to strategically design the polymers, either by genetic engineering of the producing strains or by chemical modification of the polymers. This requires a better understanding of the EPS biosynthetic pathways and their relationship with central metabolism, as well as to exploit the available polymer functionalization chemistries. Considering all this, we provide an overview of the characteristics and biological activities of cyanobacterial EPS, discuss the challenges and opportunities to improve the amount and/or characteristics of the polymers, and report the most relevant advances on the use of cyanobacterial EPS as scaffolds, coatings, and vehicles for drug delivery.

## 1. Introduction

Biopolymers are macromolecules produced by different organisms or derived from natural resources [1]. Owing to their biocompatibility, non-toxicity, flexibility, functionality, biodegradability, and possibility to be recycled by biological processes, they constitute a sustainable alternative to petrochemical-derived polymers [1,2,3]. Polysaccharides are a highly abundant and diverse group of biopolymers that can be found in all domains of life [4]. In fact, the most abundant biopolymers, cellulose and chitin, are polysaccharides [5,6]. In addition, polysaccharides display an enormous variation of physicochemical properties, such as solubility, viscosity, gelling capacity, chain length (degree of polymerization), linkage types, and patterns, which confers them the versatility to be used in a vast range of applications. Therefore, it is not surprising that they have been used for a long time in important commercial areas such as food, pharmaceuticals, biomedicine, electronics, and bioremediation [7]. 

Nowadays, the market of polysaccharides is dominated by the polymers isolated from plants (e.g., cellulose, starch, and pectins), algae (agar, alginate, carrageenan) and animals (heparin, chondroitin sulfate, hyaluronic acid), whereas those produced by bacteria still represent a small fraction of the global market [8]. However, the interest in microbial polysaccharides, particularly those of bacterial origin, is rapidly growing since they usually have shorter production times and are easier to isolate. In addition, it is possible to obtain these polymers while avoiding the use of environmentally damaging chemicals, harvesting from oceans or competition for valuable land and animal-related ethical issues [9,10]. Bacterial polysaccharide production can also occur under controlled conditions developing polymers with consistent features, whereas plant and algal polysaccharides are easily affected by climatological and geological environmental conditions [1]. Furthermore, bacterial synthesis offers an attractive alternative for the sustainable production of tailored biopolymers, reducing downstream processing [10]. Despite these advantages, the implementation of bacterial polymers in the market is hindered mainly by their high production costs [1]. Thus, the potential of these polymers is mainly in high-value market niches, such as cosmetics, pharmaceuticals, and biomedicine, in which other polymers fail to comply with the required degree of purity or lack specific functional properties. In such applications, product quality surpasses cost production and product yield issues [11]. 

Among bacteria, cyanobacteria emerge as excellent candidates for the production of polysaccharide polymers since most of the strains produce extracellular polymeric substances (EPS), mainly composed by heteropolysaccharides, with a distinctive set of industrially-desirable features including (i) strong anionic nature, (ii) presence of sulfate groups, (iii) high variety of possible structural conformations, and (iv) amphiphilic behavior [12,13]. In addition, the use of cyanobacteria as cell factories eliminates the need for carbon feedstocks, since their photoautotrophic metabolism allows a low-cost production while contributing to carbon dioxide sequestration. For this purpose, a deeper knowledge on the cyanobacterial EPS biosynthetic pathways is required, to both enhance productivity and engineer structural and compositional variants tailored for a given application.

In the last years, significant advances were made on the characterization of cyanobacterial EPS [13,14,15,16] and the validation of their biotechnological and biomedical potential as metal chelators [15,17,18,19,20,21], flocculating, emulsifying or rheology modifiers [22,23,24,25] and/or agents with valuable biological activities (e.g., antiviral, antimicrobial, anticoagulant, antitumor) [26,27,28,29,30,31,32,33,34]. However, the development of biomaterials based on these polymers has only recently started to be explored [35,36,37,38,39]. In this context, the repertoire of cyanobacterial EPS-based biomaterials can be significantly expanded through the development of strategies to obtain designer biopolymers with specialized features, by either metabolic engineering and/or chemical functionalization. Therefore, the main aim of this review is to provide the state-of-the-art on the characteristics and biological properties of cyanobacterial EPS, discuss opportunities to improve the production/characteristics of polymers by metabolic engineering, list the strategies for their extraction, purification, and functionalization, and describe the most relevant advances on the production of cyanobacterial EPS-based biomaterials.

## 2. Cyanobacterial Extracellular Polymeric Substances (EPS)

### 2.1. Polymer Characteristics 

The cyanobacterial EPS can remain attached to the cell surface as sheaths, capsules, or slimes, or be released into the surrounding environment, being designated as released polysaccharides (RPS) [12]. These polymers are heteropolysaccharides, being usually composed by a higher number of different monosaccharides (up to 13) than those produced by other bacteria, which typically contain four or fewer monomers [13]. Sugars commonly found in cyanobacterial EPS include the hexoses glucose, galactose, mannose, and fructose; the pentoses ribose, xylose, and arabinose; the deoxyhexoses fucose and rhamnose; the acidic hexoses glucuronic and galacturonic acids, the amino sugars glucosamine, galactosamine, N-acetyl galactosamine, and N-acetyl glucosamine [13]. 

Glucose is frequently the most abundant monosaccharide in cyanobacterial EPS. However, rhamnose, xylose, arabinose, fucose, mannose, and uronic acids have been found in higher amounts than glucose in some cyanobacterial polymers [40]. For example, uronic acids were the only constituents identified in the RPS of *Microcystis wesenbergii* [41], while another uncharacteristic case is that of *Cyanothece* sp. 113, which produces an extracellular polysaccharide constituted entirely by D-glucose [42]. Despite these examples, most cyanobacterial EPS have a strain-specific heterogeneous composition, which contributes to the astonishing diversity of cyanobacterial polymers. The high diversity of monosaccharidic building blocks, and consequent variety of linkages, is considered the main reason for the increased complexity and broad range of possible conformations of cyanobacterial EPS, setting them apart from other bacterial polymers [12,13]. For example, the EPS produced by *Mastigocladus laminosus* and *Cyanospira capsulata* contain repeating units of 15 monosaccharides [12]. Branching can also occur on different positions of a monosaccharide resulting in even higher structural diversity [43]. As a result, the cyanobacterial EPS usually possess a high molecular mass (in the order of MDa [14,34]) which has a direct influence on the rheological properties of the polymers [12]. The complexity of the EPS produced by cyanobacteria also make their structure elucidation challenging, and thus it is not surprising that cellulose is probably the best characterized polysaccharide in cyanobacteria [44].

Many cyanobacterial EPS also possess two different uronic acids (from 2% up to 80% of the total EPS dry weight, commonly between 15% and 30%), which is a rare feature in microbial EPS. In addition, they usually contain sulfate groups, which are usually present in the EPS produced by archaea and eukaryotic EPS but absent in those produced by bacteria. These last two features contribute to the overall anionic charge of the cyanobacterial polymers, making them suitable for a vast array of applications [12,45]. Importantly, these features are crucial for the functionalization of the polymers and contribute to their capacity to retain water and form hydrated gels [43]. The cyanobacterial EPS are also amphiphilic molecules, combining an hydrophilic fraction composed of sulfated sugars, uronic acids and ketal-linked pyruvyl groups and hydrophobic groups including ester-linked acetyl groups, deoxysugars (e.g., rhamnose and fucose) and peptidic fraction [12,13]. The presence of these hydrophobic groups strongly contributes to the emulsifying properties of polysaccharides [13]. 

Overall, the above-mentioned characteristics of these highly complex polymers make them very attractive for the biotechnological and biomedical fields.

### 2.2. Relevant Biological Activities

Initially, the research on cyanobacterial EPS was mainly focused on the potential of these polymers as bioremediation agents for the treatment of industrial and domestic wastewaters, namely for the removal of ammonia, phosphates, and heavy metals [17,18,19,20,46,47,48]. However, as knowledge accumulated, their putative antiviral, antimicrobial, antioxidant, anticoagulant, immunomodulatory, and antitumor activities started to be unveiled [26,29,30,32,33,34,49,50,51,52,53,54,55,56,57,58], opening the way for the use of cyanobacterial EPS in biomedical applications. 

Due to the limited structural information available for cyanobacterial EPS, the relationship between their structures and biological activities is far from being understood. However, the available data suggests that the negative charge and presence of sulfate groups contributes significantly to the antiviral activity displayed by several polymers [29,30,49,50,51,59]. These effects are likely due to inhibition of fusion of the enveloped virus with its target membrane, either by impairing the virus–cell attachment or the direct interaction of the negative charges of the polymer with positive charges on the virus surface [60,61]. The antiviral activity of the polymers seems to be mainly dependent on the number of negative charges and the molecular weight [60]. In the case of the sulphated polymer calcium spirulan, isolated from *Arthrospira platensis*, it was suggested that the presence of sulfate groups provides an additional contribution to the antiviral activity of these polymers by chelating calcium ions, which helps to retain the molecular conformation of the polymer [51].

The antimicrobial activity of cyanobacterial products is also well documented in the literature (reviewed in [28]). However, many of the available data were obtained using crude extracts [62,63], and thus, it is not always easy to uncouple the effects of the EPS from those resulting from the other molecules. Despite these constraints, it was demonstrated that the EPS produced by *Synechocystis* sp. R10 and *Gloecapsa* sp. Gacheva 2007/R-06/1 display antimicrobial activity against a broad spectrum of the most common food-borne pathogens [31]. Extracts of EPS released by the cyanobacterium *Arthrospira platensis* also showed antimicrobial activity against both Gram-positive and Gram-negative bacteria. Importantly, different EPS extracts showed different activities, indicating the presence of different components that differ in their solubility in the solvents employed [52]. 

A strong correlation between the sulfate content of cyanobacterial polymers and its antioxidative and anticoagulant activities was also found [26,32,33,53], and the immunomodulatory effects of specific cyanobacterial EPS were demonstrated [54]. The presence of sulfate has also been associated to the antitumor activity displayed by some EPS [34,55], although further studies are required to unveil the exact contribution of the sulfate groups. The mechanism of selective cytotoxicity displayed by different EPS with antitumor properties is also being evaluated. Studies performed with EPS isolated from *Aphanothece halophytica*, *Nostoc sphaeroides*, *Aphanizomenon flosaquae*, and *Synechocystis* ∆*sigF* revealed that the antitumor effect of these polymers is due to the induction of apoptosis in the tumor cells [27,34,57,58]. 

The vast range of biological activities displayed by cyanobacterial EPS opens a new set of possibilities for its use. However, for this process to be viable, it is necessary to complement these investigations with efforts aiming at optimizing polymer yield and tailoring its composition for specific applications.

## 3. Strategies to Optimize Cyanobacterial EPS Production and/or Polymer Characteristics

Due to their minimal nutritional requirements, cyanobacteria constitute a sustainable platform for polymer production. Moreover, depending on the environmental conditions (e.g., favorable and regular conditions), their photosynthetic metabolism allows large-scale cultivation outdoors, either in closed systems or open ponds, minimizing the costs of energy supply compared to the cultivation of e.g., heterotrophic bacteria [64]. Nevertheless, it is important to take into consideration that some cyanobacterial strains can produce toxins, and although these strains are not used for EPS production, it is essential to monitor possible contamination of cultures and/or polymers with these substances, particularly in open systems. 

Despite the advantages of using cyanobacteria for EPS production, to achieve economic viability it is necessary to optimize the production process, by (i) evaluating the best cultivation system and/or photobioreactor geometry (ii) determining the most favorable growth conditions including nutrients (carbon, macroelements, microelements), temperature, light and gases exchange, (iii) establishing of a zero-waste value chain by re-utilizing waste biomass, and (iv) optimizing downstream processing including extraction and purification of the EPS. These parameters may vary significantly depending on the strain, as already well established for the effect of the growth conditions on EPS production [12,13,15,16,65], and will not be discussed here. It is however important to emphasize that, depending on the strain, changes in the cultivation/growth conditions can affect both the amount as well as the composition of the EPS. Metabolic engineering approaches also provide an opportunity to optimize the amount of EPS produced and/or the polymers’ characteristics in order to meet industrial demands [11,66]. However, the limited information available on the cyanobacterial EPS biosynthetic process has limited the use of this approach, but the information available in the literature can provide important clues guiding future actions. 

### 3.1. Metabolic Engineering of EPS-Producing Strains

The connection between central metabolic pathways and EPS biosynthesis has been elucidated for several bacteria, opening the way for the successful optimization of EPS-producing strains such as the xanthan-producing *Xhantomonas campestris* [1,66]. More recently, the mechanisms of EPS production by cyanobacteria started to be unveiled, mainly using the model strain *Synechocystis* sp. PCC 6803 (hereafter *Synechocystis*) [67,68,69,70,71]. Nevertheless, more studies are necessary to fully understand this process in cyanobacteria. 

Studies performed in several bacteria point out that, regardless of the variety of surface polysaccharides produced, their biosynthetic pathways are relatively conserved [72]. Generally, the EPS biosynthetic pathway starts with the activation of monosaccharides and its conversion into sugar nucleotides; then, the monosaccharides are sequentially transferred from the sugar nucleotide donors to carrier molecules and assembled as repeating units. Finally, the EPS are exported to the exterior of the cell [1,72]. These steps require the participation of three groups of proteins, namely (1) enzymes involved in the biosynthesis of the sugars nucleotides, (2) glycosyltransferases to transfer the sugars to specific acceptors, and (3) proteins involved in EPS assembly, polymerization, and export [1,73,74] (Figure 1). 

All steps of the biosynthetic process offer opportunities for optimizing of the amount of EPS produced and/or its quality through genetic manipulation [11]. Here, we discuss the opportunities to improve cyanobacterial EPS production/characteristics by targeting carbon availability, synthesis of sugar nucleotide precursors, assembly of the repeating unit, and polymerization and export of the polymer.

#### 3.1.1. Carbon Availability

The production of polysaccharides is a carbon-intensive and energy-demanding process that competes with cell’s growth for available carbon resources. Thus, one of the strategies to improve EPS production consists of increasing the carbon pool of the cells, either by boosting the photosynthetic efficiency and/or the inorganic carbon intake. Previously, it was shown that the overexpression of the endogenous *Synechocystis* bicarbonate transporter BicA led to an increase in EPS production [75], and that high CO_2_ pressure boosts the generation of these polymers in *Synechococcus* sp. PCC 8806 [76]. Carbon availability can also be increased by eliminating the carbon sinks and competing pathways, such as the production of glycogen, sucrose, and compatible solutes (e.g., glucosylglycerol). The branching points between *Synechocystis’* primary metabolism and sugar nucleotide, glycogen, sucrose, and glucosylglycerol pathways is depicted in Figure 2. Glycogen is a glucose storage polymer that, in cyanobacteria, can accumulate to levels of more than 50% of the cellular dry weight, depending on the growth conditions [77]. A *Synechocystis* mutant (∆*glgC*) unable to produce glycogen possesses a higher energy charge and produces more organic acids [78]. The overexpression of the glycogen debranching enzyme GlgP, also results in massive decline of the glycogen content [79], compensating the carbon drain in an ethanol-producing *Synechocystis* mutant [79]. Although these studies unequivocally demonstrate that glycogen depletion increases the availability of carbon, it remains to be shown if this carbon surplus can be efficiently redirected towards EPS production. Regarding sucrose metabolism, the overexpression of Ugp (responsible for converting uridine triphosphate (UTP) and glucose-1-phosphate into uridine diphosphate (UDP)-glucose that serves as a substrate for sucrose and EPS synthesis) inhibited sucrose accumulation in *Synechocystis* under salt stress [80], raising the hypothesis that this effect may be due to the shift of carbon flux towards the synthesis of the exopolysaccharides [81]. A relationship between the glucosylglycerol metabolism and EPS synthesis in *Synechocystis* was also found. In this case, a mutant in a glucosylhydrolase (GghA) released higher amounts of polysaccharides (RPS) to the medium, suggesting a function of glucosylglycerol degradation via GghA in the synthesis and/or attachment of EPS to *Synechocystis* cells [82].

#### 3.1.2. Synthesis of Sugar Nucleotide Precursors

A common bottleneck in microbial EPS production is the insufficient levels of sugar nucleotides [66,74]. This aspect is particularly relevant in Gram-negative bacteria, as these precursors are also required for the production of other surface polysaccharides, including the O-antigen of the lipopolysaccharides (LPS) and the S-layer glycans [85,86]. Thus, another strategy to increase cyanoabacterial EPS production consists of increasing the levels of sugar nucleotide precursors. However, the success of this approach is still controversial [74], since it is necessary to balance the carbon supply for sugar nucleotide synthesis with glycolysis [66,74]. Higher levels of sugar nucleotides can be achieved by overexpressing enzymes such as Ugp involved in the branching-point between the cell’s primary metabolism and the sugar nucleotide pathway [10,66,74,87], as previously suggested (Figure 2) [80]. It is also necessary to consider the energetic requirements of sugar nucleotide synthesis. Availability of high-energy compounds such as adenosine triphosphate (ATP) and UTP may limit sugar nucleotide production, and therefore, strategies to increase the levels of cellular energy may also be advantageous for EPS production [66]. Finally, increasing or decreasing the synthesis of a certain type of nucleotide sugar precursor may have an impact on the EPS monosaccharidic composition [74]. Targeted modifications to obtain improved EPS for different applications are the increase in uronic acids (e.g., by targeting UDP-glucose dehydrogenase) and amino sugars (e.g., through modification of UDP-N-acetylglucosamine pyrophosphorylase). Enrichment in rare sugars such as rhamnose and fucose can also be advantageous to confer unique physical and bioactive properties to the polymers [8]. Recently, *Synechocystis*’ mutants in the tyrosine kinase Sll0923 (Wzc homologue) and/or the low molecular weight tyrosine phosphatase Slr0328 (Wzb homologue) produced EPS enriched in rhamnose [70]. Similar results had been obtained for a mutant in the ATP-binding component (Sll0982; KpsT homologue) of an EPS-related ABC transporter [68], raising the hypothesis that rhamnose metabolism is closely associated with the last steps of EPS production. This is further supported by the presence of *slr0985*, encoding a dTDP-4-dehydrorhamnose 3,5-epimerase, in close proximity to *wzc* and *kpsT* [70]. 

#### 3.1.3. Assembly of the Repeating Unit

Genetic engineering of glycosyltransferases offers a great opportunity for the optimization of the polymers’ composition and structure [74]. Overexpression of a native glycosyltransferase may increase the incorporation of the substrate sugar, provided that sufficient amounts of the sugar nucleotide are available. Alternatively, new monosaccharides may be introduced into the polymer by heterologously expressing the corresponding glycosyltransferase genes [66]. New insights into the mechanism and structure of these enzymes will enable approaches to broaden the substrate specificity and/or to swap substrate and acceptor domains from different glycosyltransferases [66,88]. However, further knowledge on this class of enzymes is necessary, as most of the cyanobacterial glycosyltransferases identified have not been characterized biochemically, making it difficult to understand their exact role in the synthesis of EPS [15]. The enzymes responsible for methylation, acetylation and pyruvylation of the EPS can also be targeted to modulate the rheological behavior of the polymers [66]. Interestingly, a *Synechocystis* mutant in a putative methyltransferase (Slr1610) displayed differences in both the molecular weight and monosaccharidic composition of its EPS compared to the wildtype [68]. Despite the significant contribution of the sulfate groups for the biological activities of the polymers, genetic engineering strategies aiming to tailor the sulfate levels in cyanobacterial EPS remain unexplored. This could be achieved by targeting the sulfotransferases responsible for the transfer of sulfate to the polymers.

#### 3.1.4. Polymerization and Export of the Polymer

A clear understanding of the last steps of EPS production and the structure/function of the proteins that participate in this process is essential to enable the rational design of engineering strategies (e.g., enzyme engineering, random mutagenesis and/or site-directed evolution) aiming at improving EPS production and/or tailoring the polymer length [10,88,89]. This last aspect is important to determine the rheological properties of the polymers as well as its potential for the production of biomaterials [66]. Therefore, targeted modification of the molecular weight by engineering the proteins involved in the polymerization, export, or degradation of the polymer (e.g., synthases, polymerases, glucosidases) represents a possibility to obtain new polymer variants [88], as successfully shown for xanthan gum and bacterial alginate [90,91].

Although the knowledge on the last steps of EPS production in cyanobacteria is limited, these mechanisms seem to be relatively conserved throughout bacteria, with the polymerization and export of the polymers usually following one of three main mechanisms: the Wzy-, ABC transporter-, or synthase-dependent pathways [88]. However, a phylum-wide analysis of cyanobacterial genomes reveled that most strains harbor gene-encoding proteins related to the three pathways but often not the complete set defining a single pathway, implying a more complex scenario than that observed for other bacteria [69]. This complexity raises the hypothesis of functional redundancy, either owing to the existence of multiple copies for some of the EPS-related genes/proteins and/or a crosstalk between the components of the different assembly and export pathways [69,70]. In agreement, mutational analyses showed that proteins related to both the Wzy- and the ABC-dependent pathways operate in *Synechocystis*’ EPS production, although their exact roles have only recently started to be elucidated [67,68,70]. Further knowledge is required to identify the bottlenecks in polymer export and pinpoint the best candidates for chain length regulation in cyanobacteria. Despite that, it was recently shown that the truncation of the C-terminal region of the *Synechocystis*’ polysaccharide copolymerase Wzc leads to an increase of the EPS attached to the cell [70] and that the deletion of a monooxigenase involved in polysaccharide degradation and recycling results in increased levels of RPS [92]. More studies are necessary to determine if these or similar modifications affect the length of the polymers obtained.

## 4. Isolation, Purification, and Functionalization of Cyanobacterial EPS 

The isolation and purification of the polymers must be cost effective, scalable, and easy to perform. It is also important to take into consideration that the methods selected influence the polymers’ yield and quality [15] and, thus, it may be necessary to adapt the protocols to the characteristics of the polymers and their final application [93]. One of the main aspects to consider is whether the EPS are attached to the cells or released to the culture medium (RPS). In the case of the EPS attached to the cells, detachment can be achieved using formaldehyde, glutaraldehyde, ethylenediaminetetraacetic acid (EDTA), sodium hydroxide, sonication, heating, cell washing with water, complexation, or ionic resins [15,16]. To select one of these methods, it is important to not only evaluate the yield, but also the levels of contamination of the polysaccharides with other cellular components. In addition, RPS are much easier to recover, being usually separated from cells by filtration and/or centrifugation. Once isolated, polymers are usually precipitated using ice cold absolute alcohols such as methanol, ethanol, or isopropanol and recovered [16,93]. The polarity of the alcohol and the low temperatures used have an impact on the yield of the polysaccharides and on the co-precipitation of impurities [16]. Despite the efficiency of selective alcohol precipitation, the costs and requirement of large amounts of precipitating agents led to the search of alternative techniques more suitable at the industrial scale, such as tangential ultrafiltration [16,94]. However, this methodology may need to be improved to minimize the problems of high viscosity of polymer solutions resulting in membrane clogging [16]. Tangential ultrafiltration can also be used to obtain a concentrated polymer solution before precipitation or spray-drying of the polymers, thus increasing the efficiency of these processes. 

After isolation of the EPS, contaminants such as inorganic salts, heavy metals, proteins, polyphenols, endotoxins, nucleic acids, or cell debris may still be present in the polymer solution. However, it is necessary to have polysaccharides with high purity levels to accurately determine their structure and composition and to obtain reproducible results for therapeutic applications [95]. Inorganic salts, monosaccharides, oligosaccharides and low molecular weight non-polar substances can be removed by dialysis. The choice of device, the molecular weight cut-off, and duration of the dialysis is very important to determine the success of this method. However, at an industrial scale, dialysis may not be a viable option. An alternative way to remove inorganic salts is through ion exchange resins, normally in the form of beads [96]. Removal of peptides and proteins can be achieved using different methods, including protease (e.g., pronase) treatment or the Sevag method (usually less efficient) [96,97]. Trichlorotrifluoroethane and trichloroacetic acids can also be used to remove proteins from the polysaccharide’s solution. However, it is necessary to consider that the first is highly volatile and, thus, has to be employed at 4 °C limiting its use, while the trichloroacetic acid is widely used but its acidity can damage the polymer structure [96,97]. The levels of polyphenol contaminants are usually reduced with charcoal washes and centrifugations, hydrogen peroxide method or functionalized resins with imidazole and pyridine [95,98]. The selection of the best purification methods depends on the characteristics of the polymers, the methods used for their isolation, and the envisaged application.

The presence of endotoxins is one of the major issues to be addressed before any biomaterial is consider safe to be used. Endotoxins are mainly due to the presence of LPS, with lipid A being responsible for most of the biological activity of these contaminants [99]. Endotoxins can significantly affect the biological effects of the polymers by eliciting a wide range of cellular responses that compromise cell viability [100,101]. Therefore, limits are imposed by regulatory entities ([102], pp. 171–175, 520–523) As an example, the food and drug administration (FDA) adopted the US Pharmacopoeia endotoxin reference standard, limiting the amount of endotoxins in eluates from medical devices to 0.5 Eurotoxin Units (EU)/mL [103]. Endotoxins are highly heat-stable and not easily destroyed by standard autoclave programs [104]. However, they can be removed by other techniques including ultrafiltration, two-phase extraction, and adsorption [99], although the efficiency of these methods depends on the characteristics of the polymer.

Depending on the application, it may be necessary to isolate fractions of the polymers with specific molecular weights. Fractionation is usually achieved by ultracentrifugation, with the added advantage of simultaneous elimination of contaminants [105]. Filtration and ultrafiltration are also popular alternatives, however, depending on the material of the filter membrane, the polysaccharides can be retained in the filter, decreasing the yield of the purification [106]. Other methods include affinity chromatography, gel chromatography, anion exchange chromatography, cellulose column chromatography, quaternary ammonium salt precipitation, graded precipitation methods, and preparative zone electrophoresis (reviewed in [96]).

The development of polysaccharide-based biomaterials often requires the chemical functionalization of the polymers. In this context, the characteristics of cyanobacterial EPS, offer a vast range of opportunities for targeted modifications (Figure 3). Successful examples of these functionalization reactions have already been described for other bacterial EPS [107,108,109,110,111,112,113,114]. The hydroxyl groups present in hexoses, pentoses, deoxyhexoses, uronic acids, and aminosugars can act as nucleophiles in base-catalyzed esterification reactions in the presence of anhydrides, esters, or carboxylic acids (Figure 3A–C). This strategy has been successfully used to fabricate photocrosslinkable hydrogels based on dextran and hyaluronic acid [108,110,114]. Another approach consists of the oxidation of diols in the presence of sodium periodate to generate reactive aldehydes, which can further react with primary amines in reductive amination reactions (Figure 3D) to produce hydrogels [107,112]. Hydroxyl groups can also undergo free radical polymerization reactions to generate graft copolymers for drug delivery (Figure 3E), as previously demonstrated for xanthan gum [109]. The carboxylic groups present in uronic acid residues allow the polymers’ functionalization through esterification or carbodiimide reactions, with the latter being particularly interesting for bioconjugation (Figure 3F) [111,113]. On the other hand, free amino groups from glucosamine residues can react with anhydrides and carboxylic acids to form amides (Figure 3G,H), or with aldehydes to form Schiff bases, which can be further reduced to imines. Overall, these chemical modifications are valuable strategies to obtain designer polymers with improved properties suitable for the development of novel biomaterials. 

For their use in biomedical applications, the polymers and/or derived biomaterials have to be biocompatible, i.e., be able to “perform with an appropriate host response in a specific application” [115]. Biocompatibility is usually evaluated in vitro by accessing the effects that biopolymers or biomaterials have on living cells [116]. Several guidelines are described in international standard protocols, with the material’s toxicity (defined as cytotoxicity) being the most common and widely used parameter evaluated (ISO 10993-5) [117]. Depending on the application, biotolerability, i.e., “the ability to reside in the body for long periods of time with only low degrees of inflammatory reaction” is an important issue to consider. This property is particularly important for non-degrading or slow-degrading implant materials [115]. Other important biosafety tests include the evaluation of the mutagenic and carcinogenic potential [118,119].

## 5. Development and Possible Applications of Cyanobacterial EPS-Based Biomaterials

Over the past few years, the development of biomaterials for therapeutic applications has become a rapidly expanding multidisciplinary field of research, with an increasing interest in uncovering novel polysaccharide-based scaffolds, coatings, and drug carriers [120]. Despite the potential of the cyanobacterial EPS and the vast range of opportunities to further improve the characteristics of the polymers by genetic engineering and/or chemical modification, the number of studies reporting their use as biomaterial is still very limited. Nevertheless, the available data represent an important step to validate the potential of cyanobacterial EPS.

The RPS produced by the cyanobacterium *Trichormus variabilis* VRUC 168 were combined with diacrylated polyethylene glycol to produce photopolymerizable hybrid hydrogels [35]. These gels were stable over time and resistant to dehydration and spontaneous hydrolysis, being successfully used as matrices for the active form of the enzyme thiosulfate:cyanide sulfur transferase, as well as for 3D culture system of human mesenchymal stem cells (hMSCs). In another study, the RPS produced by *Nostoc commune* were combined with glycerol to prepare biopolymeric films suitable for the development of new materials, including coatings and membranes [38]. Importantly, the simple and effective methodology developed allows control of the films’ thickness and mechanical properties, thus expanding the repertoire of applications in the food and biomedical industries. The polymer produced by the strong RPS producer *Cyanothece* sp. CCY 0110 [14] was also shown to be a promising vehicle for topical administration of therapeutic macromolecules. This polymer was able to spontaneously assemble with functional proteins into a new phase with gel-like behavior, and the proteins were released progressively and structurally intact near physiological conditions, primarily through the swelling of the polymer–protein matrix. The release kinetics could be modulated by the addition of divalent cations, such as calcium [37]. The same polymer combined with arabic gum was also used to generate microparticles capable of encapsulating vitamin B12 [36]. More recently, the RPS isolated from this *Cyanothece* strain was used to produce an anti-adhesive coating, obtained by spin coating (for details see [39]). This coating efficiently prevents the adhesion of relevant etiological agents, even in the presence of plasma proteins, being an important step towards the establishment of a new technological platform capable of preventing medical device-associated infections [39].

## 6. Conclusions and Future Perspectives

Owing to their characteristics and biological activities, the EPS produced by cyanobacteria are a promising platform for biotechnological and biomedical applications, including the development of novel biomaterials for therapeutic applications. However, their successful exploitation largely depends on combined efforts to optimize the amount of EPS produced and tailor their characteristics. The recent advances in the knowledge of cyanobacterial EPS biosynthetic pathways pave the way for the generation of genetically modified strains. However, there are still challenges to address, including (i) a better understanding of the relationship between central metabolism and the synthesis of sugar nucleotides, (ii) the identification and characterizing of other key components of the EPS production machinery, and (iii) elucidation of the regulatory networks of the EPS production process. Further studies, taking into account high throughput data obtained from systems biology approaches and structural information of both proteins and polymers, will be crucial to address these issues. Moving beyond cellular processes, the chemical functionalization of the polymers can also significantly increase the repertoire of cyanobacterial EPS suitable for targeted applications. The implementation of this strategy is currently limited by the lack of knowledge on the structure of cyanobacterial polymers. However, the advent of new technologies and approaches will help to overtake this bottleneck. The results obtained in the (yet limited number of) studies reporting the use of cyanobacterial EPS-based biotechnology validate their potential, encouraging future endeavors. 

## Figures and Tables

**Figure 1 ijms-20-05693-f001:**
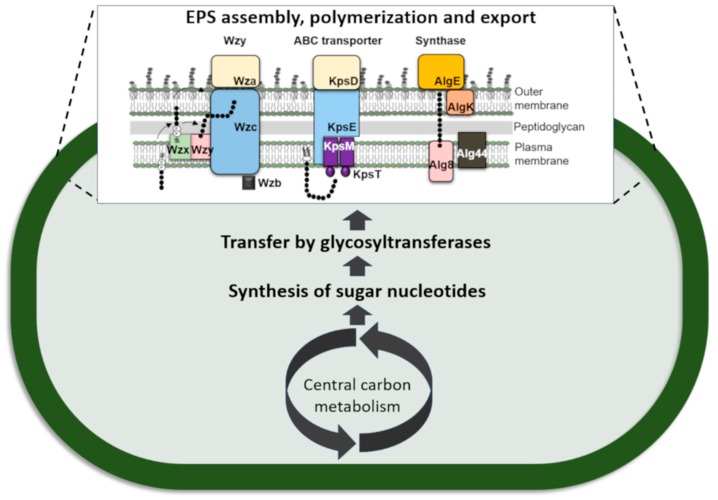
Sequence and compartmentalization of the events leading to the production of bacterial extracellular polymeric substances (EPS). EPS assembly, polymerization, and export usually follows one of three main mechanisms: the Wzy-, ABC transporter- or Synthase-dependent pathways. Adapted from [69].

**Figure 2 ijms-20-05693-f002:**
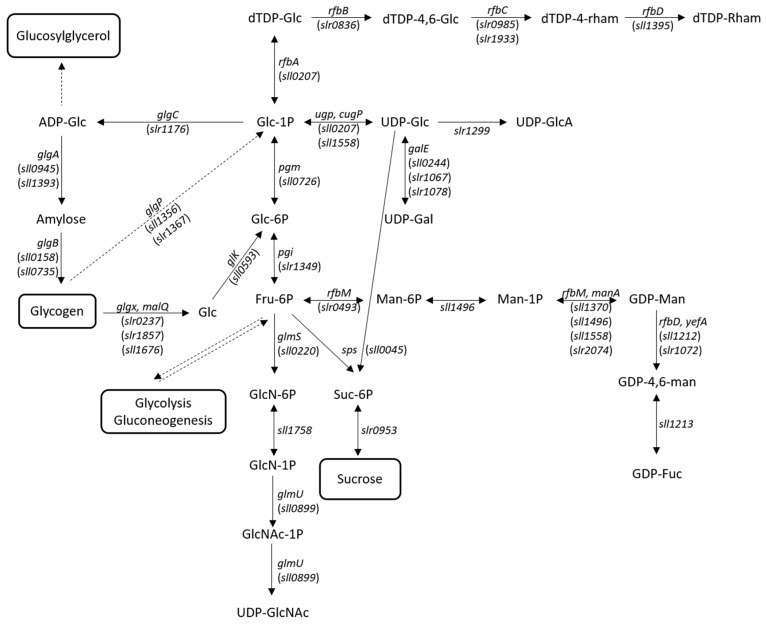
Branching points between the central carbon metabolism and the sugar nucleotide, glycogen, sucrose, and glucosylglycerol pathways in *Synechocystis* sp. PCC 6803 (based on Kegg database (https://www.genome.jp/kegg/) and [77,80,83,84]). Dotted lines indicate that the intermediary reactions are not represented. The locus tag of the genes encoding the enzymes are indicated in parenthesis. ADP-Glc: ADP-glucose; dTDP-4,6-Glc: dTDP-4-oxo-6-deoxy-D-glucose; dTDP-4-rham: dTDP-4-dehydro-beta-L-rhamnose; dTDP-Glc: dTDP-D-glucose; dTDP-rham: dTDP-L-rhamnose; Fru-6P: D-Fructose 6-phosphate; GDP-4,6-man: GDP-4-dehydro-6-deoxy-D-mannose; GDP-Fuc: GDP-L-fucose; GDP-Man: GDP-D-mannose; Glc: glucose; Glc-1P: D-glucose 1-phosphate; Glc-6P: D-glucose 6-phosphate; GlcN-1P: D-Glucosamine 1-phosphate; GlcN-6P: D-Glucosamine 6-phosphate; GlcNAc-1P: N-Acetyl-D-glucosamine 1-phosphate; Man-1P: D-Mannose 1-phosphate; Man-6P: D-Mannose 6-phosphate; Suc-6P: Sucrose 6-phosphate; UDP-Gal: UDP-D-galactose; UDP-Glc: UDP-D-glucose; UDP-GlcA: UDP-D-glucuronate; UDP-GlcNAc: UDP-N-acetyl-D-glucosamine.

**Figure 3 ijms-20-05693-f003:**
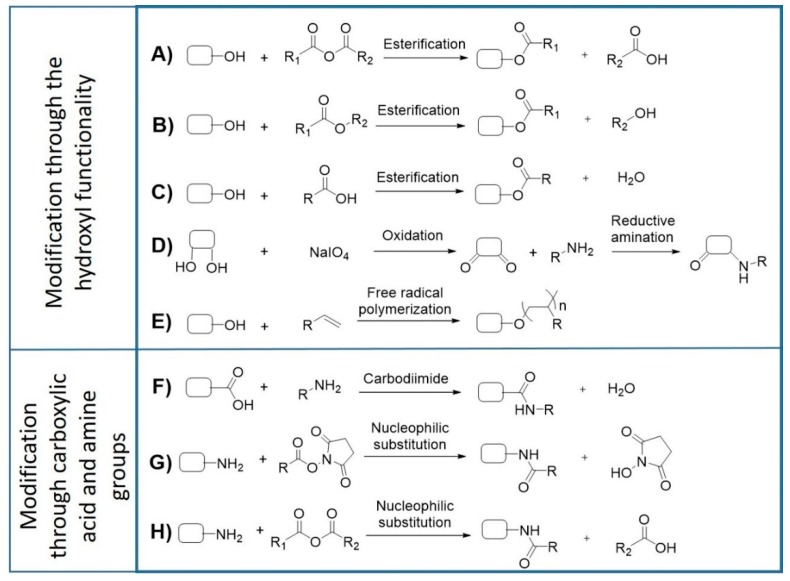
Representative strategies for functionalization of EPS by chemical modification. EPS can undergo esterification with anhydrides (**A**), esters (**B**), carboxylic acids (**C**), periodate-mediated oxidation followed by reductive amination (**D**), free radical polymerization with vinyl moiety (**E**), carbodiimide coupling (**F**) and nucleophilic substitution with esters and anhydrides (**G**,**H**), depending on the target functional group.

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
