# Peer review of "Strategies to Obtain Designer Polymers Based on Cyanobacterial Extracellular Polymeric Substances (EPS)"

_ijms, 2019, doi:10.3390/ijms20225693_

Round 1

Reviewer 1 Report

The paper "Strategies to Obtain Designer Polymers Based on Cyanobacterial Extracellular Polymeric Substances (EPS)" by Pereira et al. is a fine review about cyanobacterial EPS, an interesting field in biotechnology. Information is clearly presented, accurate and up to date.

My only concern is about English language that I think should be revised. For instance, there is an excessive employ of the article “the”, as well as of the Saxon genitive, that should be kept to the minimum. Some sentences should be rewritten, and some mistakes corrected as indicated in the revised manuscript.

Author Response

Thank you very much for reviewing our manuscript ijms-619623 "Strategies to obtain designer polymers based on cyanobacterial extracellular polymeric substances (EPS)" meant for publication in the special issue of “Designer Biopolymers: Self-Assembling Proteins and Nucleic Acids” of the International Journal of Molecular Sciences. Please find enclosed the revised version with the changes highlighted in red.

All the mistakes pointed out by the reviewer were corrected and the language was revised by an English native speaker.

We hope we have improve the manuscript into a publishable form.

Sincerely,

Sara B. Pereira

Reviewer 2 Report

This paper reviews recent challenges for designing and using extracellular polymer production by cyanobacteria. It conducts a comprehensive survey of this research field and provides state of the art knowledge. There are only a few minor point that could improve this article.

1. p 4, L 157-158. This sentence (about synthesis of metal nanoparticles) seems not to be fit for this paragraph concerning antimicrobial activity of cyanobacterial polymers.

2. p 4, L 175-176. Authors argued the merit of large scale of cultivation outdoors for minimizing the cost of energy supply. However, such approach could be easily affected by climatological and geological environmental conditions, as stated as demerit in algal polymer production (p 2, L 59-60).

3. p 11, L 443. Please state concretely how the Cyanothece RPS was developed for serving as anti-adhesive coatings. Was it modified chemically?

4. It is well-known several cyanobacteria produce varieties of toxins including nodularin, microcystins, and anatoxins. Authors may mention about the risks of handling of cyanobacterial cultures or the contamination of polymers by such toxic substances.

Author Response

Thank you very much for reviewing our manuscript ijms-619623 "Strategies to obtain designer polymers based on cyanobacterial extracellular polymeric substances (EPS)" meant for publication in the special issue of “Designer Biopolymers: Self-Assembling Proteins and Nucleic Acids” of the International Journal of Molecular Sciences. Please find enclosed the revised version with the changes highlighted in red.

1. p 4, L 157-158. This sentence (about synthesis of metal nanoparticles) seems not to be fit for this paragraph concerning antimicrobial activity of cyanobacterial polymers.

AUTHORS: We agree. The sentence was removed from this paragraph.

2. p 4, L 175-176. Authors argued the merit of large scale of cultivation outdoors for minimizing the cost of energy supply. However, such approach could be easily affected by climatological and geological environmental conditions, as stated as demerit in algal polymer production (p 2, L 59-60).

AUTHORS: Yes, it is true that outdoors cultivation of cyanobacteria for polymer production can also be affected by environmental conditions. However, depending on the cyanobacterial strain(s) and the environmental conditions at the production site (e.g. favorable and regular conditions) it allows a significant reduction of costs compared to the use of other EPS-producing bacteria. This is now clarified in the text.

3. p 11, L 443. Please state concretely how the Cyanothece RPS was developed for serving as anti-adhesive coatings. Was it modified chemically?

AUTHORS: The RPS anti-adhesive coating was obtained by spin-coating (not developed) and covalently bound to the substrate through a polydopamine (pDA) layer. This is now clarified in the text.

4. It is well-known several cyanobacteria produce varieties of toxins including nodularin, microcystins, and anatoxins. Authors may mention about the risks of handling of cyanobacterial cultures or the contamination of polymers by such toxic substances.

AUTHORS: True. A sentence mentioning the risk of culture/polymer contamination with toxins was now added to the manuscript.

We hope we have improve the manuscript into a publishable form.
Sincerely,
Sara B. Pereira